# Effective Self-supervised Pre-training on Low-compute Networks without Distillation

**Fuwen Tan**
Samsung AI Cambridge

**Fatemeh Saleh**
Microsoft Research Cambridge

**Brais Martinez**
Samsung AI Cambridge

## Abstract

Despite the impressive progress of self-supervised learning (SSL), its applicability to low-compute networks has received limited attention. Reported performance has trailed behind standard supervised pre-training by a large margin, barring self-supervised learning from making an impact on models that are deployed on device. Most prior works attribute this poor performance to the capacity bottleneck of the low-compute networks and opt to bypass the problem through the use of knowledge distillation (KD). In this work, we revisit SSL for efficient neural networks, taking a closer look at what are the detrimental factors causing the practical limitations, and whether they are intrinsic to the self-supervised low-compute setting. We find that, contrary to accepted knowledge, there is no intrinsic architectural bottleneck, we diagnose that the performance bottleneck is related to the model complexity vs regularization strength trade-off. In particular, we start by empirically observing that the use of local views can have a dramatic impact on the effectiveness of the SSL methods. This hints at view sampling being one of the performance bottlenecks for SSL on low-capacity networks. We hypothesize that the view sampling strategy for large neural networks, which requires matching views in very diverse spatial scales and contexts, is too demanding for low-capacity architectures. We systematize the design of the view sampling mechanism, leading to a new training methodology that consistently improves the performance across different SSL methods (e.g. MoCo-v2, SwAV or DINO), different low-size networks (convolution-based networks, e.g. MobileNetV2, ResNet18, ResNet34 and vision transformer, e.g. ViT-Ti), and different tasks (linear probe, object detection, instance segmentation and semi-supervised learning). Our best models establish new state-of-the-art for SSL methods on low-compute networks despite not using a KD loss term. Code is publicly available at `github.com/saic-fi/SSLight`.

## 1 Introduction

In this work, we revisit self-supervised learning (SSL) for low-compute neural networks. Previous research has shown that applying SSL methods to low-compute architectures leads to comparatively poor performance (Fang et al., 2021; Gao et al., 2022; Xu et al., 2022), i.e. there is a large performance gap between fully-supervised and self-supervised pre-training on low-compute networks. For example, the linear probe vs supervised gap of MoCo-v2 (Chen et al., 2020c) on ImageNet1K is $5.0\%$ for ResNet50 ($71.1\%$ vs $76.1\%$), while being $17.3\%$ for ResNet18 ($52.5\%$ vs $69.8\%$) (Fang et al., 2021). More importantly, while SSL pre-training for large models often exceeds supervised pre-training on a variety of downstream tasks, that is not the case for low-compute networks. Most prior works attribute the poor performance to the capacity bottleneck of the low-compute networks and resort to the use of knowledge distillation (Koohpayegani et al., 2020; Fang et al., 2021; Gao et al., 2022; Xu et al., 2022; Navaneet et al., 2021; Bhat et al., 2021). While achieving significant gains over the stand-alone SSL models, distillation-based approaches mask the problem rather than resolve it. The extra overhead of large teacher models also makes it difficult to deploy these methods in resource-restricted scenarios, e.g. on-device. In this work, we re-examine the performance of SSL low-compute pre-training, aiming to diagnose the potential bottleneck. We find that the performance gap could be largely filled by the training recipe introduced in the recent self-supervised works (Caron et al., 2020; 2021) that leverages multiple views of the images.

Comparing multiple views of the same image is the fundamental operation in the latest self-supervised models. For example, SimCLR (Chen et al., 2020a) learns to distinguish the positive and negative views by a contrastive loss. SwAV (Caron et al., 2020) learns to match the cluster assignments of

the views. We revisit the process of creating and comparing the image views in prior works and observe that the configurations for high-capacity neural networks, e.g. ResNet50/101/152 (He et al., 2016) and ViT (Dosovitskiy et al., 2021), are sub-optimal for low-capacity models as they typically lead to matching views in diverse spatial *scales* and *contexts*. For over-parameterized networks, this may not be as challenging, as verified in our experiments (sec. 4.3), but could even be considered as a manner of regularization. For lightweight networks, it results in performance degradation. This reveals a potentially overlooked issue for self-supervised learning: the trade-off between the model complexity and the regularization strength. With these findings, we further perform a systematic exploration of what aspects of the view-creation process lead to well-performing self-supervised models in the context of low-compute networks. We benchmark and ablate our new training paradigm in a variety of settings with different model architectures (MobileNet-V2 (Sandler et al., 2018), ResNet18/34/50 (He et al., 2016), ViT-S/Ti (Dosovitskiy et al., 2021)), and different self-supervised signals (MoCo-v2 (Chen et al., 2020c), SwAV (Caron et al., 2020), DINO (Caron et al., 2021)). We report results on downstream visual recognition tasks, e.g. semi-supervised visual recognition, object detection, instance segmentation. Our method outperforms the previous state-of-the-art approaches despite not relying on knowledge distillation.

Our contributions are summarized as follows:

(1) We revisit SSL for low-compute pre-training and demonstrate that, contrary to prior belief, efficient networks can learn high quality visual representations from self-supervised signals alone, rather than relying on knowledge distillation;

(2) We experimentally show that SSL low-compute pre-training can benefit from a weaker self-supervised target that learns to match *views* in more comparable spatial scales and contexts, suggesting a potentially overlooked aspect in self-supervised learning that the pretext supervision should be adaptive to the network capacity;

(3) With a systematic exploration of the view sampling mechanism, our new training recipe consistently improves multiple self-supervised learning approaches (e.g. MoCo-v2, SwAV, DINO) on a wide spectrum of low-size networks, including both convolutional neural networks (e.g. MobileNetV2, ResNet18, ResNet34) and the vision transformer (e.g. ViT-Ti), even surpassing the state-of-the-arts distillation-based approaches.

## 2 RELATED WORK

**Self-supervised learning.** The latest self-supervised models typically rely on contrastive learning, consistency regularization, and masked image modeling. Contrastive approaches learn to pull together different views of the same image (positive pairs) and push apart the ones that correspond to different images (negative pairs). In practice, these methods require a large number of negative samples. SimCLR (Chen et al., 2020a) uses negative samples coexisting in the current batch, thus requiring large batches, and MoCo (He et al., 2020) maintains a queue of negative samples and a momentum encoder to improve the consistency of the queue. Other attempts show that visual representations can be learned without discriminating samples but instead matching different views of the same image. BYOL (Grill et al., 2020) and DINO (Caron et al., 2021) start from an augmented view of an image and train the online network (a.k.a student) to predict the representation of another augmented view of the same image obtained from the target network (a.k.a teacher). In such approaches, the target (teacher) network is updated with a slow-moving average of the online (student) network. SwAV (Caron et al., 2020) introduces an online clustering-based approach that enforces the consistency between cluster assignments produced from different views of the same sample. Most recently, masked token prediction, originally developed for natural language processing, has been shown to be an effective pretext task for vision transformers. BEiT (Bao et al., 2022) adapts BERT (Devlin et al., 2019) for visual recognition by predicting the visual words (Ramesh et al., 2021) of the masked patches. SimMIM (Xie et al., 2022) extends BEiT by reconstructing the masked pixels directly. MAE (He et al., 2022) simplifies the pre-training pipeline by only encoding a small set of visible patches.

**Self-supervised learning for efficient networks.** Recent works have shown that the performance of self-supervised pre-training for low-compute network architectures trails behind standard supervised pre-training by a large margin, barring self-supervised learning from making an impact on models that are deployed on devices. One natural choice to address this problem is incorporating Knowledge

Distillation (KD) (Hinton et al., 2015) to transfer knowledge from a larger network (teacher) to the smaller architecture (student). SEED (Fang et al., 2021) and CompRess (Koohpayegani et al., 2020) transfer knowledge from the teacher in terms of similarity distributions over a dynamically maintained queue. SimReg (Navaneet et al., 2021) and DisCo Gao et al. (2022) utilize extra MLP heads to transfer the knowledge from teacher model to student model by regression. BINGO Xu et al. (2022) proposes to leverage the teacher to group similar samples into "bags", which are then used to guide the optimization of the student. While these are reasonable design choices to reduce the gap between supervised and self-supervised learning for low-compute architectures, the reason behind this gap is still poorly understood and largely unexplored. Recently, an empirical study (Shi et al., 2022) seeks to interpret the behavior of self-supervised low-compute pre-training from the perspective of over-clustering, along with examining a number of assumptions to alleviate this problem. Unlike Shi et al. (2022), we study instead from the angle of view sampling and achieve superior performance compared to the best methods with knowledge distillation.

## 3 REVISITING SELF-SUPERVISED LOW-COMPUTE TRAINING.

The main goal of this work is to diagnose the performance bottleneck of SSL methods when using lightweight networks, and to offer a solution to alleviate the problem.

### 3.1 BACKGROUND

We study representative SSL methods for the contrastive loss, MoCo-v2 (Chen et al., 2020c), clustering loss, SwAV (Caron et al., 2020) and feature matching loss, DINO (Caron et al., 2021), as they all demonstrate strong performance on large neural networks. One of the key operations within these approaches is to match multiple augmented views of the same image, involving global views that cover broader contexts (i.e. sampled from a scale range of $(0.14, 1.0)$) and local views that focus on smaller regions (i.e. sampled from a scale range of $(0.05, 0.14)$). The global and local views are typically resized to different resolutions (i.e. $224^2$ vs $96^2$). MoCo-v2 learns to align two global views of the same image while distinguishing views of different images from a queue. SwAV and DINO further make use of the local views by learning to match the cluster assignments (i.e. SwAV) or latent features (i.e. DINO) of the global and local views of an image. We start by re-examining the performance of these state-of-the-art SSL approaches in the absence of a knowledge distillation loss.

### 3.2 PILOT EXPERIMENTS

We pre-train these methods on ImageNet1K (Russakovsky et al., 2015) using MobileNetV2 (Sandler et al., 2018) as the backbone. MoCo-v2 does not use multiple crops by default. Following (Gansbeke et al., 2021), we re-implement a variant of MoCo-v2 with multiple crops, noted as MoCo-v2*. All models are trained for 200 epochs with a batch size of 1024. In Table 1, we compare the performance of the SSL methods against the supervised pre-training, as well as the distillation-based model SimReg (Navaneet et al., 2021), which is the current state-of-the-art method. For the SSL models, we use the linear evaluation protocol (He et al., 2020).

| Method | Top-1 | Top-5 |
|---|---|---|
| Supervised | 71.9 | 90.3 |
| *with KD* | | |
| SimReg | 69.1 | - |
| *two views* | | |
| MoCo-v2 | 53.8 | 77.6 |
| *multiple views* | | |
| MoCo-v2* | 60.6 | 83.3 |
| SwAV | 65.2 | 85.6 |
| DINO | 66.2 | 86.4 |

Table 1: Performance of different self-supervised methods on MobileNetV2.

**Discussion.** State-of-the-art self-supervised methods consistently underperform the supervised model and the distillation-based model by a non-negligible margin. The performance gap between MoCo-v2 and the supervised model is the largest. This is also reported in previous literature (Fang et al., 2021; Xu et al., 2022). Incorporating multiple crops largely fills the gap, improving MoCo-v2 by 6.8% in the top-1 accuracy. While using multiple crops is reported to also boost the performance of large networks, the improvement on MobileNetV2 is more significant. SwAV and DINO further reduce the self-supervised and supervised gap to 6.7% and 5.7% respectively. However, the distillation-based approach achieves a top-1 accuracy of 69.1%, which is 2.9% better than the best self-supervised model.

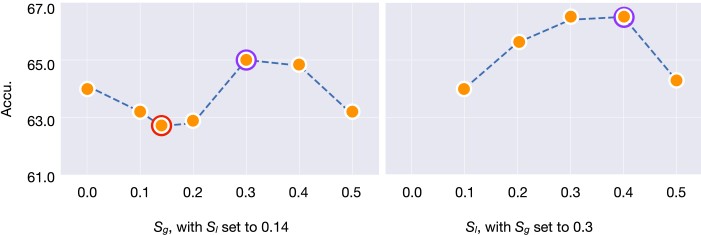

Figure 1: Experiments on the scale ranges of the image crops. **Left:** performance of different $S_g$ when $S_l$ is set to 0.14, **Right:** performance of different $S_l$ when $S_g$ is set to 0.3. The performance of the default setting is circled in Red. The best performance in each experiment is circled in Plum.

We can draw the following conclusions. For self-supervised learning on low-compute networks 1) the use of multiple views has an oversized effect and 2) learning with knowledge distillation still outperforms the best SSL method by a wide margin, showing that lightweight networks can effectively learn the self-supervised knowledge given a suitable supervisory signal. These two facts point to *the optimization signal being the cause of the performance bottleneck*.

### 3.3 MODEL COMPLEXITY VS REGULARIZATION STRENGTH TRADE-OFF: FROM THE VIEW SAMPLING PERSPECTIVE

In conventional supervised learning there is a well-known direct relation between the amount of regularization required for optimal performance and the capacity of the network (Steiner et al., 2022). Small networks require less aggressive regularization and typically perform optimally under weaker data augmentation. In self-supervised learning, the regularization itself is the optimization target. This poses the question of whether the underlying problem with self-supervised low-compute networks is not a lack of capacity, but rather a lack of modulation of the augmentation strength. In the following we study whether this is the case, concluding in the affirmative.

We investigate the problem from the view sampling perspective as it is the fundamental operation in the latest SSL methods and due to the large performance boost seen in Sec. 1 when using local views. However, given the standard parametrization of the view generation, it is difficult to disentangle the factors that make view matching challenging and, at the same time, interpretable. This is compounded by the use of local views. It is thus not clear how to design experiments with a controlled and progressive variation of the view-matching difficulty. To unearth the underlying factors, we observe that view matching becomes increasingly harder when (i) views sharing little support. This can be caused by (i.a) views representing different parts of the image (e.g. head vs legs of a dog) or (i.b) views having crop scale discrepancy (e.g. full dog vs head of dog); (ii) views having similar support, but different pixel scales (e.g. a global and a local view of a dog at $224 \times 224$ and $96 \times 96$ pixels respectively). Note here that previous research suggests that neural networks have difficulties modeling patterns at multiple *pixel scales* (Singh & Davis, 2018).

We design four different axes to explore the parametric space. In Sec.3.3.1, we focus on the relative *crop scale* to explore the impact of view support. In Sec.3.3.2, we focus on the relative *pixel scale* to explore the impact of the discrepancy in *pixel-size*. In Sec.3.3.3, we study the impact of the number of views, as more views mean higher likelihood of some pairs having good intersection and thus result in healthier supervisory signals. In Sec.3.3.4, we further examine ways to lower the impact of pairs without shared support through the modulation of the SSL loss function.

**Setting.** We use DINO as it provides the best performance in Sec. 3.2. All models are pretrained on ImageNet1K with MobileNetV2 as the backbone. We follow the same setting as in Sec. 3.2 (see also Appendix Sec. A). For the evaluation, we reserve a random subset of 50,000 images from the training set of ImageNet1K, i.e. 50 per category, and report the linear accuracy on this subset to avoid over-fitting, only using the ImageNet1K validation set in the final experiments in Sec. 4.

### 3.3.1 RELATIVE CROP SCALE

We first examine the ranges of the random scales for the global and local views. The global and local crops are created by sampling image regions from the scale ranges of $(S_g, 1.0)$ and $(0.05, S_l)$. The

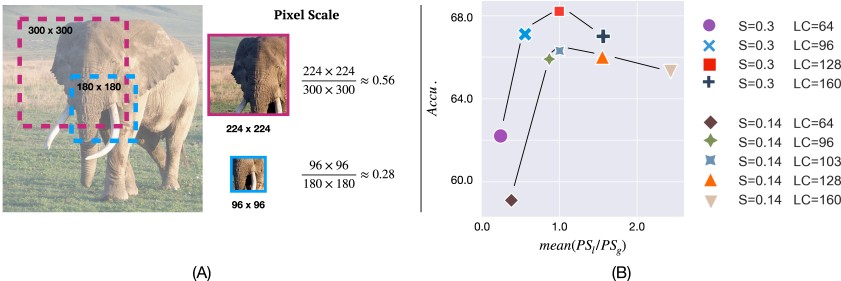

Figure 2: Left: *Pixel Scale*, i.e., the relative area between the crop in the original image and its resized area. Right: different settings and resulting performance. It is clear from the graph that optimal performance is achieved when both local and global crops have similar *PS*s ($mean(\text{PS}_g/\text{PS}_l) \approx 1.0$).

crops are then resized to 224 and 96. We hypothesize that view-matching difficulty depends on an interplay between these two scales. To explore this, we examine the impact of the scale range for the global views by varying $\text{S}_g$ while fixing $\text{S}_l$ to 0.14, the default value, and then searching $\text{S}_l$ with the obtained $\text{S}_g$. Here we decouple the search for $\text{S}_g$ and $\text{S}_l$ for simplicity.

Fig. 1 confirms that training is sensitive to the relative crop scale. Making $\text{S}_g$ too large or too small hurts performance. A larger $\text{S}_g$ reduces the variance of the global crops. This makes the matching of global views trivial and increases the discrepancy between global and local views, making their matching challenging. On the other hand, values of $\text{S}_g$ smaller than $\text{S}_l$ may lead to the mismatch in *pixel-scale* between views: the two views of the same scale are resized to different resolutions (see Sec. 3.3.2). The best performance is achieved by setting both $\text{S}_g$ and $\text{S}_l$ to $[0.3, 0.4]$.

### 3.3.2 RELATIVE PIXEL SCALE

Even if two views have similar semantic meanings, they might not correspond to the same *pixel scale*. We define the *pixel scale* of a view as $\text{PS} = \frac{\texttt{final size}}{\texttt{cropped size}}$, which can be considered an indication of the "*size of the pixel*". For each image, the global views $\{g_i\}$ and local views $\{l_i\}$ are resized to different resolutions ($\text{GC} = 224$ and $\text{LC} = 96$ respectively). Thus, the *pixel scale* of a global view $g$ ($\text{PS}_g$) and a local view $l$ ($\text{PS}_l$) are:

$$\text{PS}_g = \frac{\text{GC} \times \text{GC}}{\texttt{Area}(g)}; \qquad \text{PS}_l = \frac{\text{LC} \times \text{LC}}{\texttt{Area}(l)} \qquad (1)$$

Fig. 2(A) provides an example.

The ratio $\text{PS}_g/\text{PS}_l$ indicates the discrepancy in *pixel scale* between the global and local views. We investigate if such discrepancy has any impact on the self-supervised training. To do this, we control $\text{PS}_g/\text{PS}_l$ by varying the final resolution of the local view, $\text{LC}$. To avoid resizing views of the same scale to different resolutions, we keep $\text{S}_g = \text{S}_l = \text{S}$. We perform experiments with $\text{S}$ equals to 0.14, which is the default value, and 0.3, the optimal value we obtain in Sec.3.3.1.

Fig. 2(B) shows that matching views of different *pixel scales* results in inferior performance. Here, the optimal resolution for the local views ($\text{LC}$) is not a golden value but depends on $mean(\text{PS}_g/\text{PS}_l)$, which is the mean ratio of the *pixel scales* between the global and local views over the training set. With $\text{S}$ set to either 0.14 or 0.3, the best accuracy is achieved when $mean(\text{PS}_g/\text{PS}_l) \approx 1.0$, i.e. $\text{LC} = 103$ when $\text{S} = 0.14$, $\text{LC} = 128$ when $\text{S} = 0.3$.

### 3.3.3 NUMBER OF LOCAL VIEWS

The relative crop and pixel scales play a fundamental role, yet they are randomly sampled at every iteration. Intuitively, one could increase the number of views sampled to improve the likelihood of each image having at least some good pairs that would keep the supervisory signal healthy.

We perform experiments in different settings with different $\text{LC}$ and $\text{S}$. However, as shown in Fig.3 (left), increasing the number of local views leads to marginal improvements which also saturate quickly. We posit that randomly sampling local views may include redundant crops that provide no

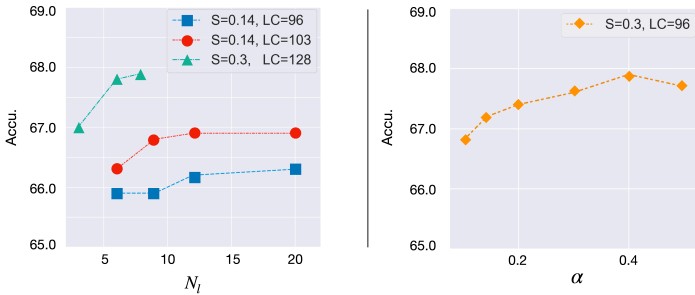

Figure 3: Left: number of local views ($x$ axis) vs performance ($y$ axis). Performance improves for more local views but saturates quickly. Right: experiments of re-balancing $\mathcal{L}_g$ and $\mathcal{L}_l$ by varying $\alpha$.

extra knowledge. Given the extra overhead of more local views and the limited benefit, we maintain the default number of views.

### 3.3.4 RE-BALANCING THE LOSS

The loss function in SSL models with local views typically includes two terms: given $N_g$ global views $\{g_i\}_{i=1}^{N_g}$ and $N_l$ local views $\{l_i\}_{i=1}^{N_l}$, the loss is computed as:

$$\mathcal{L} = \frac{\mathcal{L}_g + \mathcal{L}_l}{P_{gg} + P_{gl}} \tag{2}$$

$\mathcal{L}_g = \sum_{i \neq j} \text{SL}(g_i, g_j)$ aggregates losses between global views (i.e. *global-global* pairs), $\mathcal{L}_l = \sum_{i,j} \text{SL}(g_i, l_j)$ aggregates losses between global and local views (i.e. *global-local* pairs), $\text{SL}$ is the self-supervised loss, which could be contrastive loss, cluster loss, or feature matching loss, etc, $P_{gg} = N_g \times (N_g - 1)$ is the number of *global-global* pairs, whereas $P_{gl} = N_g \times N_l$ is the number of *global-local* pairs. As $N_g$ is usually smaller than $N_l$, $\frac{P_{gg}}{P_{gl}} \ll 1$. For example, with the default $N_g = 2$, $N_l = 6$, $\frac{P_{gg}}{P_{gl}} = 0.167$. In other words, the loss $\mathcal{L}$ is dominated by $\mathcal{L}_l$. On the other hand, aligning *global-local* pairs is also more difficult than aligning *global-global* pairs, especially when the network capacity is low. Therefore, we re-balance the loss by re-weighting $\mathcal{L}_g$ and $\mathcal{L}_l$:

$$\mathcal{L} = \alpha \cdot \frac{\mathcal{L}_g}{P_{gg}} + (1 - \alpha) \cdot \frac{\mathcal{L}_l}{P_{gl}} \tag{3}$$

Here $\alpha$ is a hyper-parameter that re-weights the contributions of $\mathcal{L}_g$ and $\mathcal{L}_l$. Note that with the default numbers of views, the original formulation is equivalent to setting $\alpha$ to $0.143$. Fig.3 (right) depicts that re-balancing $L_g$ and $L_l$ improves the performance. The optimal $\alpha$ falls in the range $[0.3, 0.4]$. We believe using $\alpha$ further helps modulate the crucial trade-off between the strength of the self-supervision and the model capacity.

With the findings discussed, our best pre-trained model with MobileNetV2 achieves a top-1 linear accuracy of $68.3\%$ on the ImageNet1K validation set, a gain of $2.1\%$ over the DINO model, as shown in Table 2. Note that the exploration is not exhaustive as different axes may correlate. We conjecture that a larger scale search in the design space could lead to further improvement.

| | DINO | DINO with new crop scales | DINO with new crop&pixel scales | Ours |
|---|---|---|---|---|
| Top-1(%) | 66.2 | 67.0 (+0.7) | 67.9 (+1.7) | 68.3 (+2.1) |

Table 2: Experiments on the ImageNet1K validation set.

## 4 EVALUATION

We perform evaluations across three dimensions; i) different self-supervised methods; ii) different low-compute architectures; iii) different downstream tasks, and further compare our pre-trained

model to the current state-of-the-art methods. Our experiments follow the benchmarks set forth by prior works on the topic. However, we also include results for ViT-Ti and ViT-S (Dosovitskiy et al., 2021), which while they have not been covered in prior work we believe offer an important data point. We re-use the best view sampling recipe from Sec. 3.3, found for DINO with MobileNetV2 through linear probing on ImageNet1K, and apply it to all architectures and SSL methods. We remark that the positive and consistent results show conclusively the generality of the findings. Dedicated studies for different models/tasks could potentially lead to further improvement.

## 4.1 IMPLEMENTATION

All models are pre-trained on ImageNet1K (Russakovsky et al., 2015) for 200 epochs. Following Caron et al. (2021), we use the LARS optimizer (You et al., 2017) for the convolution-based networks, and AdamW (Loshchilov & Hutter, 2019) for the vision transformers. We use a batch size of 1024 and a linear learning rate warm-up in the first 10 epochs. After the warm-up, the learning rate decays with a cosine schedule (Loshchilov & Hutter, 2017). Most of the other hyper-parameters are inherited from the original literature. We provide further details in Appendix Sec. A.

## 4.2 EXPERIMENTS ON DIFFERENT SELF-SUPERVISED METHODS.

We start by verifying if the new training paradigm could be transferred to different self-supervised approaches. Here we perform experiments on representative methods as discussed in Sec.3.2: MoCo-v2* (Chen et al., 2020c), SwAV (Caron et al., 2020), and DINO (Caron et al., 2021). The visual backbone used in these experiments is MobileNetV2. As reported in Table 3, all self-supervised methods consistently improve under the proposed training regime, confirming the findings extend to SSL approaches that rely on contrastive learning, clustering, and feature matching. The performance gap between the self-supervised and fully-supervised methods is reduced to 3.6% in the top-1 accuracy, and 2.5% in the top-5 accuracy. Note that the models presented here are pre-trained for only 200

| Method | Linear eval. on ImageNet-1K | |
| | Top-1(%) | Top-5(%) |
| --- | --- | --- |
| Supervised | 71.9 | 90.3 |
| MoCo-v2* | | |
| Baseline | 60.6 | 83.3 |
| Ours | 61.6 (+1.0) | 84.2 (+0.9) |
| SwAV | | |
| Baseline | 65.2 | 85.6 |
| Ours | 67.3 (+2.1) | 87.2 (+1.6) |
| DINO | | |
| Baseline | 66.2 | 86.4 |
| Ours | 68.3 (+2.1) | 87.8 (+1.4) |

Table 3: Experiments on MoCo-v2, SwAV, DINO with MobileNetV2. The green numbers depict the improvement over the baseline model.

epochs with a batch size of 1024, while the state-of-the-art self-supervised training is shown to perform best on larger settings, e.g. 800 epochs with a batch size of 4096. We conjecture the self-supervised and fully-supervised gap could be further reduced by scaling up the experiments.

## 4.3 EXPERIMENTS ON DIFFERENT LOW-COMPUTE ARCHITECTURES.

We perform experiments on other lightweight architectures that were widely evaluated in prior literature (Koohpayegani et al., 2020; Fang et al., 2021; Gao et al., 2022; Xu et al., 2022). We include both the convolution-based networks, e.g. ResNet18 and ResNet34 as well as the vision transformers, e.g. ViT-Ti (Dosovitskiy et al., 2021), and use DINO as the base model. Results of medium-capacity networks, ResNet50 and ViT-S/16 (Dosovitskiy et al., 2021), are also included. For the medium-capacity networks, we use a batch size of 640 instead of 1024 due to hardware limitations. Table 4 and 5 report the evaluations under the linear probe protocol. Our pre-trained models consistently outperform the DINO baseline on all low-compute networks: a gain of 3.5%, 2.0%, 4.7%, 2.8% in top-1 for ResNet18, ResNet34, ViT-Ti/32, and ViT-Ti/16. It shows that the low-compute networks can all benefit from the new view-sampling recipe. On the other hand, our model performs on par with DINO on ResNet50. For ViT-S/16, the improvement also drops to 0.7%. From Fig. 4, we can also observe that *the smaller the network the larger the improvement*. These results confirm our hypothesis that the best SSL performance is achieved by modulating the regularization in accordance to the network capacity.

| Backbone | Linear eval. on ImageNet-1k | | | | | | | |
|---|---|---|---|---|---|---|---|---|
| | Top-1(%) | Top-5(%) | Top-1(%) | Top-5(%) | Top-1(%) | Top-5(%) | Top-1(%) | Top-5(%) |
| | MobileNetV2 (#Par.2.2M; GFLOPS 0.31) | | ResNet18 (#Par.11.2M; GFLOPS 1.8) | | ResNet34 (#Par.21.3M; GFLOPS 3.67) | | ResNet50 (#Par.23.5M; GFLOPS 4.12) | |
| Supervised | 71.9 | 90.3 | 69.8 | 89.1 | 73.3 | 91.4 | 76.1 | 92.9 |
| DINO | 66.2 | 86.4 | 62.2 | 84.0 | 67.7 | 88.4 | 73.4 | 91.5 |
| Ours | 68.3 (+2.1) | 87.8 (+1.4) | 65.7 (+3.5) | 86.6 (+2.6) | 69.7 (+2.0) | 89.5 (+1.1) | 73.4 (+0.0) | 91.5 (+0.0) |

Table 4: Experiments on the Convolutional Networks using DINO (Caron et al., 2021) as the base model. The green numbers depict the improvement over the DINO baseline.

| Backbone | Linear eval. on ImageNet-1k | |
|---|---|---|
| | Top-1(%) | Top-5(%) |
| | ViT-Ti/32 (#Par.5.5M; GFLOPS 0.31) | |
| Supervised | - | - |
| DINO | 55.4 | 78.1 |
| Ours | 60.1 (+4.7) | 81.8 (+3.7) |
| | ViT-Ti/16 (#Par.5.5M; GFLOPS 1.26) | |
| Supervised | 72.2 | 91.1 |
| DINO | 66.7 | 86.6 |
| Ours | 69.5 (+2.8) | 88.5 (+1.9) |
| | ViT-S/16 (#Par.21.7M; GFLOPS 4.61) | |
| Supervised | 79.9 | 95.0 |
| DINO | 75.4 | 92.3 |
| Ours | 76.1 (+0.7) | 92.7 (+0.4) |

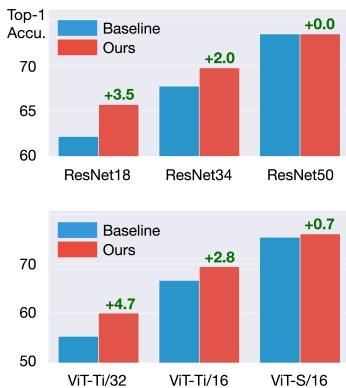

Table 5: Experiments on the Vision Transformers. The supervised results are from DeiT (Touvron et al., 2021).

Figure 4: Improvements vs model complexity. The improvement of our model decreases when the network architecture becomes larger.

## 4.4 FINETUNING FOR OBJECT DETECTION/INSTANCE SEGMENTATION.

We demonstrate the transfer capacity of our pre-trained models on the COCO object detection and instance segmentation task (Lin et al., 2014). Following He et al. (2020), we train the Mask R-CNN FPN model (He et al., 2017) using the pre-trained MobileNetV2, ResNet18, ResNet34 as the feature

backbones. We re-use all the hyperparameters in He et al. (2020). The models are finetuned for $1\times$ schedule (i.e. 12 epochs) by SGD (Loshchilov & Hutter, 2017) on an 8-GPU server. We train the models on the COCO 2017 training set (117k samples) and report the mean average precision score (mAP@100) on the COCO 2017 validation set (5000 samples).

As shown in Table 6, our pre-trained models achieve consistent improvements over DINO on all lightweight backbones for both the box and mask predictions. Similar to the evaluation on ImageNet1K (sec. 4.3), we observe *larger improvements on smaller backbones*. Compared to the fully-supervised backbone, our pretrained ResNet18/34 models perform comparably in general, even slightly better on the mask prediction task: $+0.5\%$ AP$^{mk}$ for ResNet34.

| Backbone | Object Detection | | | Instance Segmentation | | |
|---|---|---|---|---|---|---|
| | AP$^{bb}$ | AP$^{bb}_{50}$ | AP$^{bb}_{75}$ | AP$^{mk}$ | AP$^{mk}_{50}$ | AP$^{mk}_{75}$ |
| *MobileNetV2* | | | | | | |
| Supervised | 33.1 | 52.9 | 35.5 | 29.8 | 49.5 | 31.0 |
| DINO | 30.9 | 51.3 | 32.1 | 28.1 | 47.6 | 29.1 |
| Ours | 32.1 | 52.5 | 33.8 | 29.1 | 48.8 | 30.2 |
| | (+1.2) | (+1.2) | (+1.7) | (+1.0) | (+1.2) | (+1.1) |
| *ResNet18* | | | | | | |
| Supervised | 34.5 | 54.7 | 37.4 | 31.6 | 51.6 | 33.6 |
| DINO | 32.7 | 53.5 | 34.8 | 30.6 | 50.4 | 32.3 |
| Ours | 34.1 | 54.8 | 36.3 | 31.8 | 51.8 | 33.8 |
| | (+1.4) | (+1.3) | (+1.5) | (+1.2) | (+1.4) | (+1.5) |
| *ResNet34* | | | | | | |
| Supervised | 38.7 | 59.1 | 42.4 | 35.0 | 56.0 | 37.4 |
| DINO | 37.6 | 58.7 | 40.8 | 34.6 | 55.6 | 36.8 |
| Ours | 38.6 | 59.9 | 41.9 | 35.5 | 56.9 | 37.9 |
| | (+1.0) | (+1.2) | (+1.1) | (+0.9) | (+1.3) | (+1.1) |

Table 6: Experiments on COCO object detection/instance segmentation. AP$^{bb}_*$/AP$^{mk}_*$ are the mean average precisions of the bounding boxes/instance masks. Numbers in green indicate the performance gain compared to the DINO baseline.

| Method | Teacher | | Training | MobileNetV2 | | ResNet18 | | ResNet34 | |
| | Model | Epoch | Epoch | Top-1 | Top-5 | Top-1 | Top-5 | Top-1 | Top-5 |
|---|---|---|---|---|---|---|---|---|---|
| Supervised | - | - | - | 71.9 | 90.3 | 69.8 | 89.1 | 73.3 | 91.4 |
| Shi et al. (2022) | - | - | 800 | - | - | 55.7 | - | - | - |
| ReSSL | - | - | 200 | - | - | 58.1 | - | - | - |
| CompRess | MoCoV2 R50 | 800 | 130 | 65.8 | - | 62.6 | - | - | - |
| SimReg | MoCoV2 R50 | 800 | 130 | **69.1** | - | 65.1 | - | - | - |
| SEED | MoCoV2 R152 | 400 | 200 | - | - | 59.5 | 83.3 | 62.7 | 85.8 |
| DisCo | MoCoV2 R152 | 400 | 200 | - | - | 65.5 | 86.7 | 68.1 | 88.6 |
| BINGO | MoCoV2 R152 | 400 | 200 | - | - | 65.9 | 87.1 | 69.1 | 88.9 |
| SEED | SWAV R50x2 | 800 | 400 | - | - | 63.0 | 84.9 | 65.7 | 86.8 |
| DisCo | SWAV R50x2 | 800 | 200 | - | - | 65.2 | 86.8 | 67.6 | 88.6 |
| BINGO | SWAV R50x2 | 800 | 200 | - | - | 65.5 | 87.0 | 68.9 | 89.0 |
| Ours | - | - | 200 | 68.3 (-0.8) | 87.8 | 65.7 (-0.2) | 86.6 (-0.5) | 69.7 (+0.6) | 89.5 (+0.5) |
| | - | - | 400 | 68.8 (-0.3) | 88.3 | **66.8** (+0.9) | **87.3** (+0.2) | **70.8** (+1.7) | **90.0** (+1.0) |

Table 7: Comparison to the state-of-the-art methods on ImageNet1K under the linear evaluation protocol. The best number is in **bold**. The best number from the *baseline* models is underlined. The green/red text indicates the performance gain or gap compared to the best state-of-the-art model.

## 4.5 COMPARISON TO THE STATE-OF-THE-ART ON IMAGENET1K

In Table 7, we benchmark our pre-trained models with the state-of-the-art low-compute pretraining works on ImageNet1K, under the linear probe protocol. We compare to both the standalone self-supervised methods, e.g. Shi et al. (2022) and ReSSL (Zheng et al., 2021), as well as the KD based methods, e.g. CompRess (Koohpayegani et al., 2020), SimReg (Navaneet et al., 2021), SEED (Fang et al., 2021), DisCo (Gao et al., 2022), and BINGO (Xu et al., 2022).

Our models perform significantly better than the standalone self-supervised approaches, e.g. $+10.0\%/+7.6\%$ compared to Shi et al. (2022) and ReSSL on ResNet18. When pre-trained for similar numbers of epochs, our models perform on par with the best KD-based models on all the low-compute backbones, e.g. $-0.8\%/-0.2\%/+0.6\%$ in the top-1 accuracy for MobileNetV2/ResNet18/ResNet34. Note that these latest approaches all rely on knowledge distillation (KD), where the teacher models (i.e. ResNet50/50x2/152) are significantly larger than the target low-compute network (i.e. MobileNetV2) and are pre-trained in significantly larger settings (batch size of 4096 for 400/800 epochs). On the other hand, our models are pre-trained *from scratch* in a smaller setting (batch size of 1024 for 200 epochs) without using any distillation signals. When pre-trained for 400 epochs with the same batch size (1024), the performance of our models is further improved, even surpassing the state-of-the-art methods on ResNet18 ($+0.9\%$ in Top-1) and ResNet34 ($+1.7\%$ in Top-1). We believe these results show the feasibility of applying self-supervised pretraining on low-compute networks.

On COCO, our model outperforms the best competitor when using MobileNetV2 by $1.7\%$ in mAP for object detection, and by $1.6\%$ for object segmentation. While Sec. 4.4 showed that the gains of the proposed method also transfer to downstream tasks, the current results show that it actually *transfers better* than that of competing methods. This is the first time that an SSL method on low-compute networks roughly bridges the gap with supervised pre-training for downstream tasks. Complete object detection and semantic segmentation results on COCO are shown in the Appendix Sec. C.

We provide comparisons to the state-of-the-art methods on semi-supervised learning on ImageNet1K. Our model outperforms the best existing approach with ResNet18 by $1.3\%$ and $2.5\%$ for $1\%$ and $10\%$ labeled data respectively. Again, more complete results are included in the Appendix Sec. B.

## 5 CONCLUSION

In this paper, we investigate the feasibility of self-supervised low-compute pre-training, seeking to diagnose the performance bottleneck reported in previous literature. We reveal the previously overlooked problem of the trade-off between the model complexity and the regularization strength for self-supervised learning and demonstrate the importance of the design choices for view sampling. We show that by learning with proper image views, self-supervised models could achieve strong performance without knowledge distillation. We hope that these encouraging results could motivate more interesting research on efficient representation learning for low-compute models/devices.

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

## A    FURTHER DETAILS OF THE EXPERIMENTAL SETTINGS

We provide further details of the pre-training and linear evaluation experiments discussed in 3, and 4. Table 8 presents the implementation details shared by all the experiments covering different self-supervised approaches (i.e. MoCo-v2, SwAV, DINO) and different network backbones (i.e. MobileNetV2, ResNet18/34/50, ViT-Ti/S). Note that the learning rates of the pretraining for different SSL approaches are different, depending on the pretext tasks. Particularly, we use 0.48 for MoCo-v2 and DINO, 0.3 for SwAV. With LARS as the optimizer, the depth-wise convolution layers and bias terms are excluded from the layer-wise LR scaling. All the other settings, including the intensity-based data augmentations, are inherited from the original literature. Code for the pre-training and evaluation is publicly available at `github.com/saic-fi/SSLight`.

| Pre-training config. | CNNs | | ViTs | |
|---|---|---|---|---|
| | MobileNetV2 ResNet18/34 | ResNet50 | ViT-Ti/32 ViT-Ti/16 | ViT-S/16 |
| *Multiview sampling* | | | | |
| $S_g$, $S_l$ (Sec.3.3.1) | 0.3 | 0.3 | 0.3 | 0.3 |
| GC/LC (Sec.3.3.2) | 224/128 | 224/128 | 224/128 | 224/128 |
| $N_l$ (Sec.3.3.3) | 6 | 6 | 10 | 10 |
| $\alpha$ (Sec.3.3.4) | 0.4 | 0.4 | 0.4 | 0.4 |
| *Pre-training* | | | | |
| optimizer | LARS | LARS | AdamW | AdamW |
| optimizer momentum | 0.9 | 0.9 | $\beta_1, \beta_2{=}0.9, 0.999$ | $\beta_1, \beta_2{=}0.9, 0.999$ |
| training epochs | 200 | 200 | 200 | 200 |
| warmup epochs | 10 | 10 | 10 | 10 |
| batch size | 1024 | 640 | 1024 | 640 |
| base learning rate (*lr*) | $\text{lr} \times \frac{\text{batchsize}}{256}$ | $\text{lr} \times \frac{\text{batchsize}}{256}$ | $5\text{e-}4 \times \frac{\text{batchsize}}{256}$ | $5\text{e-}4 \times \frac{\text{batchsize}}{256}$ |
| minimum learning rate | $\text{lr} \times 1\text{e-}3$ | $\text{lr} \times 1\text{e-}3$ | 1e-5 | 1e-5 |
| base weight decay (*wd*) | 1e-6 | 1e-6 | 0.04 | 0.04 |
| minimum weight decay | 1e-6 | 1e-6 | 0.4 | 0.4 |
| *lr/wd* schedule | cosine decay | cosine decay | cosine decay | cosine decay |
| *Linear probing* | | | | |
| optimizer | LARS | LARS | SGD | SGD |
| optimizer momentum | 0.9 | 0.9 | 0.9 | 0.9 |
| training epochs | 100 | 100 | 100 | 100 |
| warmup epochs | 0 | 0 | 0 | 0 |
| batch size | 4096 | 4096 | 4096 | 4096 |
| base learning rate (*lr*) | $0.05 \times \frac{\text{batchsize}}{256}$ | $0.05 \times \frac{\text{batchsize}}{256}$ | $0.01 \times \frac{\text{batchsize}}{256}$ | $0.01 \times \frac{\text{batchsize}}{256}$ |
| minimum learning rate | 1e-6 | 1e-6 | 1e-5 | 1e-5 |
| weight decay | 0 | 0 | 0 | 0 |
| *lr* schedule | cosine decay | cosine decay | cosine decay | cosine decay |

Table 8: Implementation details for the pretraining and evaluation experiments discussed in 3, and 4.

## B    SEMI-SUPERVISED EVALUATION ON IMAGENET1K

Following CompRress (Koohpayegani et al., 2020), SEED (Fang et al., 2021), DisCo (Gao et al., 2022), and BINGO (Xu et al., 2022), we perform semi-supervised evaluations on ImageNet1K (Russakovsky et al., 2015) by finetuning the pre-trained models on the 1% and 10% labeled data defined in SimCLR Chen et al. (2020a)[1]. Recall that for linear probing, the classifier is applied on top of the spatially pooled features of the convolutional networks. For the semi-supervised evaluation however, we apply the classifier on the second layer of the MLP head (Caron et al., 2021)), similar to SimCLR (Chen et al., 2020a;b)), PAWS (Assran et al., 2021). We find this significantly improves the classification accuracy under a low-data/label regime. During training, only random cropping and horizontal flipping are used for pre-processing, the images are then resized to $224 \times 224$. During inference, the images are first resized to have a minimum edge of 256 then cropped in the center to $224 \times 224$. All models are optimized by the SGD (w. Nesterov) optimizer with a batch size of 1024 and a momentum of 0.9. The initial learning rate is set to $0.03 \times \text{batchsize}/256$ and decay to 1e-6 by a cosine schedule (Loshchilov & Hutter, 2017). For the experiment with 10% labeled data, we finetune for 30 epochs. For the experiment with 1% labeled data, we finetune for 60 epochs.

---

[1] `https://www.tensorflow.org/datasets/catalog/imagenet2012_subset`

| Backbone | # Pretrained Epochs | 1% labels | | 10% labels | |
|---|---|---|---|---|---|
| | | Top-1(%) | Top-5(%) | Top-1(%) | Top-5(%) |
| MobileNetV2 (#Par.2.2M; GFLOPS 0.31) | | | | | |
| DINO | 200 | 47.9 | 75.1 | 61.3 | 84.2 |
| Ours | 200 | 50.6 (+2.7) | 77.6 (+2.5) | 63.5 (+2.2) | 85.8 (+1.6) |
| Ours | 400 | 52.6 (+4.7) | 78.6 (+3.5) | 64.0 (+2.7) | 85.8 (+1.6) |
| ResNet18 (#Par.11.2M; GFLOPS 1.8) | | | | | |
| DINO | 200 | 44.5 | 72.1 | 59.2 | 82.8 |
| Ours | 200 | 49.8 (+5.3) | 77.2 (+5.1) | 63.0 (+3.8) | 85.5 (+2.7) |
| Ours | 400 | 51.6 (+7.1) | 78.5 (+6.4) | 63.7 (+4.5) | 86.0 (+3.2) |
| ResNet34 (#Par.21.3M; GFLOPS 3.67) | | | | | |
| DINO | 200 | 52.4 | 79.0 | 65.4 | 86.9 |
| Ours | 200 | 55.2 (+2.8) | 81.5 (+2.5) | 67.2 (+1.8) | 88.4 (+1.5) |
| Ours | 400 | 56.8 (+4.4) | 82.6 (+3.6) | 67.5 (+2.1) | 88.5 (+1.6) |

Table 9: Semi-supervised evaluation on ImageNet1K, with MobileNetV2, ResNet18, ResNet34 as backbones. The green numbers depict the improvement over the baseline model.

| Method | Teacher Model (Accu.) | Epoch | Pretrained Epoch | 1% labels | 10% labels |
|---|---|---|---|---|---|
| SEED Fang et al. (2021) | R152 (74.1) | 400 | 200 | 44.3 | 54.8 |
| DisCo Gao et al. (2022) | R152 (74.1) | 400 | 200 | 47.1 | 54.7 |
| BINGO Xu et al. (2022) | R152 (74.1) | 400 | 200 | 50.3 | 61.2 |
| BINGO Xu et al. (2022) | R50x2 (77.3) | 800 | 200 | 48.2 | 60.2 |
| Ours | - | - | 200 | 49.8 (-0.5) | 63.0 (+1.8) |
| | - | - | 400 | **51.6** (+1.3) | **63.7** (+2.5) |

Table 10: Comparison to the state-of-the-art methods on semi-supervised ImageNet1K recognition, using ResNet18 as the backbone. The best number is in **bold**. The best number from the *baseline* models is underlined. The green/red text indicates the performance gain or gap compared to the best state-of-the-art model.

Table 9 shows that our pre-training paradigm significantly improves the semi-supervised performance in all settings. For example, with 1% labeled data, we improve the DINO baseline (Caron et al., 2021) by 2.8%/5.3%/2.7% on MobileNetV2/ResNet18/ResNet34. Note that the performance gain in semi-supervised learning is larger than that in linear probing, demonstrating further advantages of our model under low-data/label regime.

In Table 10, we compare to the state-of-the-art approaches on semi-supervised ImageNet1K recognition, using ResNet18 as the backbone. Recall that the baseline models are all optimized by knowledge distillation. Our models are pre-trained *from scratch* without using any teacher. With the same number of pre-trained epochs (i.e. 200 epochs), our model performs on par with (i.e. -0.5) the best baseline for 1% labeled data, favorably (i.e. +1.8) for 10% labeled data. With more pre-trained epochs (i.e. 400 epochs), our model outperforms the best state-of-the-art method by +1.3/+2.5 for 1%/10% labeled data. Note that, compared to the large-scale pre-training of the teacher models used in the baselines, the overhead of the extra epochs for our model is marginal.

## C  EXTRA EXPERIMENTS ON COCO

We compare our model to the state-of-the-art self-supervised low-compute methods on COCO Instance Segmentation(Lin et al. (2014)). While similar evaluations have been reported in previous literature (Fang et al., 2021; Gao et al., 2022; Xu et al., 2022), the detail configuration is not made public. For fair comparison, we use the pre-trained weights released by the baseline approaches (Navaneet et al., 2021; Fang et al., 2021; Gao

| Method | Mask R-CNN FPN on MobileNet-V2 | | | | | |
|--------|------------|--------------|--------------|------------|--------------|--------------|
| | $AP^{bb}$ | $AP^{bb}_{50}$ | $AP^{bb}_{75}$ | $AP^{mk}$ | $AP^{mk}_{50}$ | $AP^{mk}_{75}$ |
| | $1\times$ | | | | | |
| SimReg | 30.7 | 49.4 | 32.7 | 27.7 | 46.3 | 28.7 |
| Ours | **32.4** (+1.7) | **52.4** (+3.0) | **34.4** (+1.7) | **29.3** (+1.6) | **49.1** (+2.8) | **30.3** (+1.6) |
| | $2\times$ | | | | | |
| SimReg | 33.7 | 52.8 | 36.1 | 30.0 | 49.3 | 31.3 |
| Ours | **35.1** (+1.4) | **55.4** (+2.6) | **37.8** (+1.7) | **31.7** (+1.7) | **52.1** (+2.8) | **33.3** (+2.0) |

Table 11: Comparison to the state-of-the-art method, SimReg (Navaneet et al., 2021), on COCO Instance Segmentation, using MobileNet-V2 as the backbone. The best number is in **bold**. The green numbers depict the improvement over the SimReg model.

| Method | Teacher | Mask R-CNN FPN on ResNet18 | | | | | |
|--------|---------|------------|--------------|--------------|------------|--------------|--------------|
| | | $AP^{bb}$ | $AP^{bb}_{50}$ | $AP^{bb}_{75}$ | $AP^{mk}$ | $AP^{mk}_{50}$ | $AP^{mk}_{75}$ |
| | | $1\times$ | | | | | |
| SimReg | BYOL R50 | 33.1 | 52.1 | 36.1 | 30.2 | 49.2 | 32.1 |
| SEED | SWAV R50x2 | **34.7** | 54.8 | **37.5** | 31.9 | 51.8 | 34.1 |
| DisCo | SWAV R50x2 | 34.6 | 55.1 | 37.1 | 32.0 | 52.0 | 34.0 |
| Ours | - | **34.7** (+0.0) | **55.6** (+0.5) | **37.5**(+0.0) | **32.3** (+0.3) | **52.8** (+0.8) | **34.3** (+0.2) |
| | | $2\times$ | | | | | |
| SimReg | BYOL R50 | 35.7 | 55.6 | 38.9 | 32.6 | 52.5 | 35.1 |
| SEED | SWAV R50x2 | 36.9 | 57.2 | 40.3 | 33.6 | 54.3 | 35.8 |
| DisCo | SWAV R50x2 | 36.6 | 56.9 | 39.7 | 33.6 | 54.0 | 35.9 |
| Ours | - | **37.3** (+0.4) | **58.2** (+1.0) | **40.7** (+0.4) | **34.3** (+0.7) | **55.1** (+0.8) | **36.8** (+0.9) |

Table 12: Comparison to the state-of-the-art methods on COCO Instance Segmentation, using ResNet18 as the backbone. The best number is in **bold**. The best number from the *baseline* models is underlined. The green text indicates the performance gain compared to the best state-of-the-art model.

et al., 2022) and finetune all the models in the Mask R-CNN FPN framework(He et al. (2017)), using the default configuration defined in Detectron2 (Wu et al., 2019). Our models used in this experiment are pre-trained for 400 epochs. All models are evaluated under both the $1\times$ and $2\times$ schedules. For the MobileNetV2 backbone that is not supported in Detectron2(Wu et al. (2019)), we re-implement the Torchvision Instance Segmentation framework[2] in Detectron2(Wu et al. (2019)).

Table 11 shows the comparison to the SimReg model (Navaneet et al., 2021) on COCO Instance Segmentation, using MobileNetV2 as the backbone. We use the best model released by the SimReg authors, which is trained with MoCo-v2 R50 (Chen et al., 2020c) as the teacher and achieves a top-1 accuracy of 69.1 under linear evaluation (see Sec.4.5 in the main paper). Our models consistently outperform SimReg in all settings.

In Table 12, we further compare our model to the state-of-the-art approaches using ResNet18 as the backbone. Note that the SimReg baseline used in this experiment is pre-trained with BYOL R50 (Grill et al., 2020) as the teacher. It is shown to be the best model for ResNet18. Our model demonstrates better performance over the best state-of-the-art method in all settings, especially for the mask prediction task. However, the improvements are smaller (i.e. $0.0\% \sim 1.0\%$) than the linear/semi-supervised tasks in general. We conjecture that, with more in-domain data/labels as in COCO, the advantage of good pre-trained models becomes marginal.

---

[2]https://pytorch.org/vision/stable/models.html#object-detection-instance-segmentation-and-person-keypoint-detection

