# OpenReview forum: "Effective Self-supervised Pre-training on Low-compute Networks without Distillation"
_ICLR.cc/2023/Conference — ICLR 2023 poster_

### Official Review · Reviewer_aQkw · 2022-10-21

**Confidence:** 3
**Correctness:** 3
**Technical Novelty And Significance:** 3
**Empirical Novelty And Significance:** 3
**Recommendation:** 8

**Clarity, Quality, Novelty And Reproducibility:**

Most part of the paper is written clearly. There are some small confusions, as follows:
1. The concepts of global vs local views are mentioned early but not well defined until when defining pixel scales. I would suggest the authors give some background on $S_g, S_{\ell}$ shortly before introducing the pilot experiments. Assuming such knowledge from the readers would limit the range of the audience.
2. Many numbers of accuracies, gaps, and hyper-parameters are mentioned in the paragraphs in Sec 3, it is possibly better to absorb them in compact tables and refer to them rather than listing these numbers directly, which slow downs reading quite a bit. The same happens in later sections where the authors constantly mention experiment details, which require careful reading to understand. By summarizing their intuitions rather than listing specific details, the paragraphs would look better and reader-friendly.


**Strength And Weaknesses:**

Strengths:
1. The research goal is well-motivated. The wider performance gap between supervised and self-supervised methods on smaller models is long observed and dubbed the benefits of larger model sizes. However, as many applications of deep learning are still confined by memory and compute limits, it is important to improve these methods in smaller models.
2. The proposed method in the paper is motivated due to their careful experiments and observations. The authors conduct controlled experiments supporting their intuitions about the correlations of model sizes and data augmentation strength in SSL. They define a rather novel measure pixel scale to examine the effects of different view construction on the performances. And based on these results, they conjectured the smaller performance gap between SSL and supervised for larger models is due to that the smaller models lack the ability to learn on such adversarial augmented data.
3.  The comparison between different methods on different downstream tasks are solid and convincing. The author compared their improved SSL method with others on classification, object detection, and segmentation, and over multiple architectures including resnet, mobilenetV2 and small transformers. The performance looks good and surpasses distillation-based methods in many cases, which is clearly an important contribution.

Weaknesses:
1. There are some clarity issues due to arrangement of contents, which are mentioned below.
2. The authors can dig into the local vs global augmentations with more principled metrics. The $\mathrm{mean}(PS_g/PS_{\ell})$ seems rather arbitrary and less intuitive.


**Summary Of The Paper:**

This paper studies the problem of improving self-supervised learning (SSL) methods on compact neural networks. The authors investigate the relations between data augmentation strength and sizes of neural networks in SSL, and propose a new objective and data augmentation scheme to improve the performance.

**Summary Of The Review:**

This paper studies an important but less-explored aspect of self-supervised pretraining. They investigate the problem with interesting experiments to support intuitions and provide motivations, and their final results justify their contributions.

---

> ### Author Response · Authors · 2022-11-09
> **Response to Reviewer aQkw**
>
> We thank the reviewer for reading our paper in detail, for the clear and constructive feedback, and for the highly positive assessment.
>
> ***
>
> Q4.1: Clarity
>
> A4.1: Thanks for the constructive comment. We agree these changes would help the wider audience. In the revised paper, we will include the necessary background on global and local views as suggested and use tables to summarize the experimental settings, and use the main body of text to discuss results/intuitions.
>
> ***
>
> Q4.2: The authors can dig into the local vs global augmentations with more principled metrics. The mean(PSg/PSℓ) seems rather arbitrary and less intuitive.
>
> A4.2: We’d like to further clarify our motivation: PSg/PSℓ represents the relative pixel scales between local and global views (intuitively, how “blurry” the views are). Prior research [1,2] has shown that neural networks have difficulties learning representations invariant to such value. Thus, when PSg/PSℓ approaches 1.0, we would expect this task to be easier, and gradually become harder when PSg/PSℓ departs from 1.0. However, due to the random nature of the view sampling process, we need to encode some statistical measures like the mean, leading to mean(PSg/PSℓ) as a measure of how challenging the view creation is.
>
> We agree that there is definitely more room to explore augmentations, including which “axis of variation” are explored. We observe however that the performance gap between supervised and unsupervised learning is already approaching that seen for larger models, potentially indicating performance is saturating.
>
> [1] Multi-scale Context Aggregation by Dilated Convolutions (ICLR 2016)
>
> [2] An Analysis of Scale Invariance in Object Detection - SNIP (CVPR 2018)

---

### Official Review · Reviewer_MNWJ · 2022-10-25

**Confidence:** 4
**Correctness:** 3
**Technical Novelty And Significance:** 2
**Empirical Novelty And Significance:** 2
**Recommendation:** 6

**Clarity, Quality, Novelty And Reproducibility:**

In general, the writing of this paper is clear. However, I have some concerns on the novelty and importance of this paper. See the weaknesses above.

**Strength And Weaknesses:**

Strength:

- The experimental analysis in Section 3 is extensive.
- The improvements in Table 3 are impressive.

Weaknesses:

- The authors find that a weaker training target is beneficial for the self-supervised learning of low-compute networks, and demonstrate that this idea can be implemented by adjusting the relative scale of different views. Personally, I do not think this is a significant scientific contribution. The conclusion seems straightforward, while the authors conduct a series of experiments, yielding an empirical principle for hyper-parameter searching. In general, I think that, currently, this paper may be too engineering-oriented to be published on ICLR.
- Is the proposed method compatible with knowledge distillation? For low-compute networks, the most significant concern may be how to improve the accuracy without introducing additional computational costs during inference. I think that "without distillation" is not an important advantage.



**Summary Of The Paper:**

This paper proposes a self-supervised learning approach in the context of low-compute deep-learning models (e.g., DeiT-Tiny). The authors observe that a weaker self-supervised target is beneficial for small networks, and propose to match multiple views in more comparable spatial scales and contexts. Experimental results are provided on ImageNet and COCO.

**Summary Of The Review:**

The experimental analysis is extensive. However, I have some concerns on the novelty and importance of this paper. See the weaknesses above.

---

> ### Author Response · Authors · 2022-11-15
> **Response to Reviewer MNWJ**
>
> We thank the reviewer for the comments and answer the questions in order:
>
> ***
>
> Q3.1: Lack of novelty. The conclusion is straightforward. The paper is too engineering-oriented.
>
> A3.1: We believe the sub-optimal performance of SSL on low-compute networks is an important problem and deserves a thorough investigation. There has been a consensus in attributing this issue to a capacity bottleneck when using an SSL loss, see [1,2,3,4,5,6,7]. Therefore, we consider our work a *scientific contribution* regarding the conclusion.
>
> Also, kindly note that prior art, spanning several years, show a consensus on the cause of this low performance [1,2,3,4,5,6,7]. including a diagnostics study [5]. In light of this pervasive misunderstanding, we believe our conclusion is not *straightforward*.
>
> We conduct a large range of experiments in order to *support the conclusions*. We thus believe the claims of being too engineering-oriented come from devoiding the experiments from their context: we conduct the experiments to support the main claim. We also see value in our experiments achieving state-of-the-art performance, outperforming prior works on standard benchmarks, with a simplified method.
>
> ***
>
> Q3.2: Is our method compatible with knowledge distillation?
>
> A3.2: Prior work developed KD-based approaches under assumptions that do not hold in our case. We agree that possible future research would be how to further improve the performance of a well-performing SSL objective through Knowledge Distillation on top. However, this is also a full research topic. We believe a single data point without proper in-depth investigation would not be meaningful.
>
> ***
>
> Q3.3: “For low-compute networks, the most significant concern may be how to improve the **accuracy** without introducing additional computational **costs during inference**.”.
>
> A3.3: We agree with the reviewer’s comment. Our results achieve state-of-the-art across standard benchmarks, as defined by prior work: ImageNet classification (linear probe), object detection, instance segmentation, and semi-supervised learning, without introducing additional costs during inference. In order to provide extra support, we go beyond prior benchmarks and also provide results with the transformer architectures.
>
> That said, we consider that lowering the complexity of the training pipeline and reducing the training overhead are still friendly and practical advantages of our work.
>
> ***
>
> Q3.4: "Without distillation" is not an important advantage.
>
> A3.4: The consensus among previous research is that performance degradation is due to the capacity bottleneck. Based on this observation, the community has focused on two major approaches: [1,2,3,4] opted to bypass the problem via knowledge distillation, while [6,7] searched for new architectures tailored to SSL. We believe our work is *impactful* as it reveals this widely accepted assumption is incorrect. To perform the diagnosis, we first need to remove the distillation loss.
>
> ***
>
> [1] CompRess: Self-Supervised Learning by Compressing Representations. NeurIPS 2020.
>
> [2] SEED: Self-supervised Distillation For Visual Representation. ICLR 2021.
>
> [3] Bag of Instances Aggregation Boosts Self-supervised Distillation. ICLR 2022.
>
> [4] DisCo: Remedy Self-supervised Learning on Lightweight Models with Distilled Contrastive Learning. ECCV 2022.
>
> [5] On the Efficacy of Small Self-Supervised Contrastive Models without Distillation Signals. AAAI 2022.
>
> [6] DSPNet: Towards Slimmable Pretrained Networks based on Discriminative Self-supervised Learning. Arxiv 2022.
>
> [7] Slimmable Networks for Contrastive Self-supervised Learning. Arxiv 2022.

---

> > ### Comment · Reviewer_MNWJ · 2022-12-04
> > **Updating the score**
> >
> > After reading both the rebuttal of the authors and the comments from other reviewers, I'm convinced that the contributions of this paper are not as engineering-oriented as I have thought. The value of this paper may lie in it reveals that "low-compute networks can also be effectively trained with self-supervised algorithms". However, I still believe it is important to demonstrate that the proposed method is compatible with knowledge distillation. Based on these comments, I vote for "borderline acceptance".

---

### Official Review · Reviewer_rtAP · 2022-10-25

**Confidence:** 4
**Correctness:** 3
**Technical Novelty And Significance:** 2
**Empirical Novelty And Significance:** 3
**Recommendation:** 5

**Clarity, Quality, Novelty And Reproducibility:**

The experiments overall are well-designed, while the technical novelty is limited.

**Strength And Weaknesses:**

Strength:
- This paper is well-motivated. The proposed method provides a good alternative to the current distillation based methods for ssl pretraining on low-compute networks.
- The article clearly shows the thought process and is easy to follow.
- The proposed method shows decent improvement over baseline results.

Weakness:
- I'm not convinced by how S_l and S_g are produced. 0.14 and 0.4 for S_l are quite different and I'm wondering whether it will influence the conclusion.
- Lack of analyzes. Most parts of the method are only driven by empirical results.

**Summary Of The Paper:**

In this paper, the authors investigate the problem of self-supervised pre-training on compact networks. They study the impact of view sampling strategy and design a strategy which can boost the SSL pretraining performance without knowledge distillation. The developed method consistently shows improvement on different compact models with different SSL frameworks.

**Summary Of The Review:**

In conclusion, this work improves the self-supervised pretraining on compact models by tuning several hyperparameters in the view sampling strategy, which is simple but shows decent performance boost. From my perspective, I would like to see more analyzes on how these changes influence the training, instead of just showing the empirical results.

---

> ### Author Response · Authors · 2022-11-09
> **Response to Reviewer rtAP**
>
> We thank the reviewer for the comments. We answer the questions in order.
>
> ***
>
> Q2.1: About S_I and S_g. 0.14 and 0.4 for S_l are quite different and whether it will influence the conclusion.
>
> A2.1: If we understand correctly: the **difference** mentioned by the reviewer refers to the performance gap between different S_l (0.1 vs 0.4) with S_g=0.3 shown in Figure 1, and how it compares to that in Table 3. This results in concern by the reviewer that designs other than S_g and S_l (e.g. pixel scale, re-balancing) are not beneficial.
>
> If this is the correct interpretation, we would like to note that the evaluations of Figure 1 and Table 3 were performed on **different** subsets of ImageNet (see 4th paragraph of Sec. 3.2). Table 1 uses a reserved training subset, while the official validation set is used for Table 3. We use a reserved training set to avoid overfitting to the final test set (ImageNet val) during the exploration.
>
> i) on the reserved training set, we believe the designs other than S_g and S_l have been demonstrated to be effective in Figure 2 and 3 of the main paper.
>
> ii) on the official ImageNet validation set, we provide an ablation showing the top1 linear accuracy of three models, with MobileNetV2 as the backbone:
>
> || DINO baseline (S_g = S_l = 0.14) | DINO w S_g = S_l = 0.3   | Our final model |
> |----------|-------------|-------------|-------------|
> | top1 accu. | 66.2        | 67.0 (+0.8)       | 68.3 (+2.1)        |
> ***
>
> We will include this Table with the ablation on the official ImageNet validation set in the revised paper.
>
> If we have misinterpreted the question please let us know and we will do our best to reply accordingly.
>
> ***
>
> Q2.2: Lack of analysis.
>
> A2.2: We would be grateful if the reviewer could kindly suggest what type of extra analysis we can run to improve the content of the paper.
>
> In this paper, we conjecture that the self-supervised signal for large networks is too strong for light-weight networks. We further derive several concrete hypotheses from the view sampling perspective (Sec. 3.2). These hypotheses are intuitive and empirically justified by our study (i.e. >= 38 reported experiments). We consider this hypothesis+justification pipeline as an effective analysis.
>
> Also, as SSL models are typically trained by pretext tasks, we believe evaluating on downstream tasks is the most direct and effective way to validate SSL models, which we do thoroughly.
>
> That being said, we would be happy to improve our paper by further expanding the analysis to other aspects of our SSL models.

---

### Official Review · Reviewer_h6ge · 2022-11-02

**Confidence:** 3
**Correctness:** 2
**Technical Novelty And Significance:** 2
**Empirical Novelty And Significance:** 2
**Recommendation:** 8

**Clarity, Quality, Novelty And Reproducibility:**

*Clarity* : The paper is clearly written, figures and presentation are easy to follow

*Evaluation Quality* : The paper seems quite reproducible, with a relatively thorough ablation section providing key information about reproducing their results.

**Strength And Weaknesses:**

**Strengths**
1. The paper has extensive ablations that validate the impact of various modifications to the classical training procedure for vision-based SSL
2. We see non-trivial improvements of the method across multiple benchmarks and on top of multiple SSL methods.

**Weaknesses**

My main issue with the paper is that it is not properly positioned in that I do not believe it has provided adequate evidence for its major claim.  The paper claims to show that excessive regularization from the augmentation strategy is the reason why smaller models under-perform KD  with larger models.
What the authors present however, is a set of  very useful **training recipes** for getting improved performance on smaller models. The existence and efficacy of this recipe however does not support their claim that capacity is not a bottleneck for smaller models. Specifically, to fully show this they would have to :
1. Demonstrate that the present training recipe **does not improve results for larger models**. If it does lead to improved results for the larger model, then the authors have only presented a better training procedure but not an explanation for the performance gap. Improved results on a larger model roughly implies improved KD performance which would mean that the purported gap isn't actually closed but the baselines are just shifted up.

Table 5 buttresses my suspicion that the procedure presented will benefit models at larger scales, given that the delta improvement actually **increases** from Resnet18 to Resnet34

[Edit - ] The authors rightly pointed out that Table 5 is an unfair comparison to their method. They directed me to table 3, which I agree is the better table to compare to.
I have updated my score based on the rebuttal

**Summary Of The Paper:**

This paper presents  a training recipe for helping smaller computer vision models achieved improved performance from Self-Supervised Learning. The paper presents the success of said recipe as a evidence that it is not the capacity of the smaller models that limits their performance but rather the extreme regularization effects of some of the data augmentation procedures used in classical SSL training.

**Summary Of The Review:**

The authors do a good job of presenting a training recipe that improves performance of SSL methods on smaller models.
However, they do not establish the key claim they make in the paper which is that
> We find that, contrary to accepted knowledge, there is no intrinsic architectural bottleneck, we diagnose that the performance bottleneck is related to the model complexity vs regularization strength trade-off

For my score to increase, I would have to be convinced that the positioning of the paper is correct.

[Edit - ] The authors rightly pointed out that Table 5 is an unfair comparison to their method. They directed me to table 3, which I agree is the better table to compare to.
I have updated my score based on the rebuttal

---

> ### Author Response · Authors · 2022-11-09
> **Response to Reviewer h6ge**
>
> We thank the reviewer for the comments, and in particular for recognizing “the extensive ablations” and “the non-trivial improvements”.
>
> We would like to note that our paper is not simply a training recipe, but also a diagnosis of SSL for low-compute networks. We believe this by itself is a very valuable contribution. There is a widely accepted view in the community that lack of network capacity is the core issue that prevents SSL from working on low-compute networks. This assumption has driven the research for the past years, with works following one of two paths: rely on KD (and thus tweak the KD recipe) and re-design the architecture with SSL in mind (e.g. NAS).
> We strongly believe that a core contribution of our paper is to challenge these widely accepted assumptions, hopefully contributing to steering future research directions.
>
> Q1.1: Lack of adequate evidence for the major claim about the capacity vs regularization trade-off.
>
> A1.1: We agree with the reviewer that we need to show that **“the present training recipe does not improve results for larger models.”** We indeed included such results in our paper.
>
> Table 5 does not directly support the claim as it presents the comparison between our models and the state-of-the-art KD-based methods. The deltas show the improvements of our models over the **best KD-based** baselines for **different** network architectures. Therefore, the delta cannot be interpreted as solely coming from the training recipe.
>
> An apples-to-apples comparison is however shown in Table 3, where we show the performance *with* and *without* our method. For completeness, we copy-pasted these results below.
>
> Convolutional networks:
>
> |Arch. (GFLOPS)| MobileNetV2 (0.31) | ResNet18 (1.8)    | ResNet34 (3.67)    | ResNet50 (4.12)   |
> |----------|-------------|-------------|-------------|-------------|
> | Baseline | 66.2        | 62.2        | 67.7        | 73.4        |
> | Ours     | 68.3 (+2.1) | 65.7 (+3.5) | 69.7 (+2.0) | 73.4 (+0.0) |
>
> As shown in the table, and also as elaborated in the **second paragraph of Sec. 4.3**: *“For the ResNet-like architectures, we observe that the smaller the network the larger the improvement.”* and that *“our model performs on par with DINO on ResNet50.”*
>
> Vision Transformers:
>
> | Arch. (GFLOPS) | ViT-Ti/32 (0.31) | ViT-Ti/16 (1.26) | ViT-S/16 (4.61) |
> |----------------|------------------|------------------|-----------------|
> | Baseline       | 55.4             | 66.7             | 75.4            |
> | Ours           | 60.1 (+4.7)      | 69.5 (+2.8)      | 76.1 (+0.7)     |
>
>
> For Vision Transformers (ViTs), we included the results of ViT-Ti/16 and ViT-S/16 in table 3. To offer further evidence, we also include in this response the result for ViT-Ti/32 (ViT-Ti with a patch size of 32 ). We make a similar observation that *the smaller the network, the larger the improvement.*  For a medium-size network like ViT-S/16, the gap is reduced to 0.7%. We will add the ViT-Ti/32 results to the main paper.
>
> While we would be happier if the training recipe applied across all capacities, the experiments show that it is not the case. We believe these results offer great support to that claim and fulfill the condition laid out by the reviewer to accept the core paper hypothesis.

---

> > ### Comment · Reviewer_h6ge · 2022-11-09
> > **Updated Score - thanks for the response**
> >
> > Thank you for the clarification.
> > I have updated my score.
> > I think it might be best to present Table 3 as a graph, showing the gap between the methods as the model size increases.
> > This would make it much easier to parse.

---

### Author Response · Authors · 2022-11-18
**Summary of the revision**

Dear Reviewers/AC,

We are grateful to all reviewers for their comments and helpful feedback to improve our work! Based on the feedback from the reviewers, we uploaded a revised draft. The key revisions are as follows:

i) We included the background of the view samping process for previous SSL methods (Sec.3.1) (Reviewer aQkw)

ii) We summarized the implementation details in a table (Appendix Sec. A) (Reviewer aQkw)

iii) We added Fig.4, which shows the performance gap between our model and the baseline as the network size increases (Reviewer h6ge)

iv) We added Table 2, which presents the ablations on the ImageNet1K validation set with DINO+MobileNetV2 (Reviewer rtAP)

As always, please let us know if you have any further questions/concerns.

Thanks!

---

### Decision · Program_Chairs · 2023-01-20

**Decision:**

Accept: poster

**Justification For Why Not Higher Score:**

Interesting, but not ground breaking kind of work

**Justification For Why Not Lower Score:**

Training small models with self-supervised learning is a timely and challenging problem. This work provides a new angle and approach to tackle the problem without using knowledge distillation.

**Metareview: Summary, Strengths And Weaknesses:**

The paper aims to find a way to self-train low-compute networks effectively without knowledge distillation. The most significant finding according to the work is that the unfavourable performance of low-compute networks in the self-supervised learning framework, is attributed to the model complexity versus regularization trade-off during self-supervised learning. Through careful design of training strategies, the authors conclude that it is feasible to obtain state-of-the-art performance for low-compute networks without knowledge distillation. This is in contrast to the common understanding. The claim is supported by experiments. I agree with the reviewers to accept the paper. (In the paper, "a closer at" should read "a closer look at".)

**Note From Pc:**

if the above contains the word "oral" or "spotlight" please see: "oral" presentation means -> notable-top-5% and "spotlight" means -> notable-top-25%. As stated in our emails, we are disassociating presentation type from AC recommendations